# Effects of water and fertilizer coupling on the physiological characteristics and growth of rabbiteye blueberry

Xiaolan Guo[1], Shuangshuang Li[2], Delu Wang[1]*, Zongsheng Huang[3], Naeem Sarwar[4], Khuram Mubeen[5], Muhammad Shakeel[6], Mubshar Hussain[4]

1 College of Forestry, Guizhou University, Huaxi, Guiyang, Guizhou, China, 2 Forestry Bureau of Xixiu District, Anshun City, Guizhou, China, 3 School of Architecture and Urban Planning, Guizhou University, Huaxi, Guiyang, Guizhou, China, 4 Faculty of Agricultural Sciences and Technology, Department of Agronomy, Bahauddin Zakariya University Multan, Multan, Pakistan, 5 Department of Agronomy, MNS University of Agriculture Multan, Multan, Pakistan, 6 Provincial Key Laboratory for Agricultural Pest Management of Mountainous Regions, Institute of Entomology, Guizhou University, Guiyang, Guizhou, China

Ⓔ These authors contributed equally to this work.

* deluwang23@aliyun.com

**Data Availability Statement:** All relevant data are within the manuscript and its Supporting Information files.

## Abstract

Understanding the impact of irrigation and fertilizer on rabbiteye blueberry (*Vaccinium virgatum*) physiology is necessary for its precision planting. Here, we applied varied irrigation and fertilizer under completely randomized experimental design to see its impact on the physiological characteristics and bush growth of rabbiteye blueberries. A comprehensive evaluation of the membership function was used to establish the best water–fertilizer coupling regimes. Rabbiteye blueberry enhanced the net photosynthetic rate, stomatal conductance and transpiration rate of leaf and improved its photosynthetic capacity at maximum level of irrigation water and fertilizer application (F3W4). The high fertilizer–medium water treatment (F3W3) increased leaf-soluble protein contents. The medium fertilizer–medium water treatment (F2W3, F2W2) increased leaf- soluble sugar, superoxide dismutase, and chlorophyll contents; decreased the malondialdehyde content; and enhanced leaf resistance and metabolism. It also promoted the growth of flower buds and new shoots. Combined membership function and cluster analyses revealed that the optimal water and fertilizer conditions for promoting rabbiteye blueberry plant growth were the medium fertilizer–medium water [$(NH_4)_2SO_4$:$Ca(H_2PO_4)_2$:$K_2SO_4$ at 59:10:20 g plant$^{-1}$; 2.5 L water plant$^{-1}$], medium fertilizer–medium-high water [$(NH_4)_2SO_4$:$Ca(H_2PO_4)_2$:$K_2SO_4$ at 59:10:20 g plant$^{-1}$; 3.75 L water plant$^{-1}$], and high fertilizer–medium-high water [$(NH_4)_2SO_4$:$Ca(H_2PO_4)_2$:$K_2SO_4$ at 118:20:40 g plant$^{-1}$; 3.75 L water plant$^{-1}$] treatments. The findings of this study could be used in improving the precision and efficacy of rabbiteye blueberry planting in Guizhou, China. Such an approach can increase the productivity and profitability for local fruit farmers.

**Funding:** Author thanks the National Natural Science Foundation of China (NSFC 31260192) for their financial support.

**Competing interests:** The authors do not have any conflicts of interest.

# 1 Introduction

Moisture and nutrients are the two most important factors affecting plant growth and agricultural production, which can be readily controlled by farmers. Water and fertilizers can have interactive effects on plant growth and nutrient-utilization efficiency [1]. Therefore, optimized fertilizer and water coupling is pre-requisite to get higher resources use efficiencies along with higher growth, fruit yield and quality [2,3].

Water shortages and sub-optimal fertilizer application restrict crop yields and agricultural production. Plants exposed to drought stress undergo oxidative damage due to over generation of reactive oxygen species (ROS); these ROS are highly toxic and damage photosynthetic machinery of plants [4]. Nonetheless, under water deficit conditions, plants are unable to uptake applied nutrients leading to reduced resource use efficiency [4]. Plants have an adaptive mechanism to counteract the damaging effects of ROS by over generation of anti-oxidant enzymes like super oxide dismutase (SOD), peroxidases and catalase [4,5]. Drought tolerant plants have higher anti-oxidant potential than sensitive ones [4,5]. Nevertheless, excess water and fertilizer are also often applied, wasting limited water resources and causing environmental pollution [6]. This is particularly problematic in arid regions with impoverished soil such as in the karst area of Guizhou, China. Zotarelli et al. [7,8] found that nitrogen application did not increase tomato (*Lycopersicon esculentum* Mill.) yield, whereas irrigation markedly enhanced the utilization efficiency of nitrogen fertilizers. Moreover, the interaction between water and fertilizer boosted the yield by 11–80% compared to conventional control methods, but had no effect on soil solute migration. In another study, Hochmuth et al. [9] reported that nitrogen application at a rate $>224$ kg ha$^{-1}$ failed to further increase tomato fruit yield and lowered the fertilizer utilization efficiency up to 87%; because when fertilizer application rates are high, large amounts of nitrates accumulated in the soil. In their water and fertilizer coupling experiments on jujube (*Ziziphus jujuba*), Liu et al. [10] reported that the interaction between the water and fertilizer substantially increased fruit yield as well as water and nitrogen utilization efficiency. Therefore, investigations into water and fertilizer coupling have important theoretical and practical implications in terms of designing high-efficiency, high-yield crop irrigation and fertilization systems that conserve limited resources, and are cost-effective and sustainable.

Blueberry is an emerging cosmopolitan fruit tree and a commercially important bush crop in North America. The name blueberry refers to the blue fruit produced by member species within the genus *Vaccinium* (Ericaceae). Rabbiteye blueberry (*Vaccinium corymbosum*) is a small species native to North America. It was introduced to China in the mid- to late 1980s and, in recent years, has been promoted on a trial basis in various provinces and cities of Southern China. Blueberry cultivation has helped to alleviate local poverty and enhance the regional economy. During 2017, Rabbiteye blueberry was the dominant species cultivated in China, as it was cultivated on 31,210 ha$^{-12}$ with total production of 114,905 tons. However, only in Guizhou province this crop occupies area of 13,000 ha$^{-1}$ with total production of 30,000 tons fruit during 2017 [11].

Blueberries are shrubs with shallow, inconspicuous main roots, few root hairs, and low drought tolerance [12]. Therefore, for blueberry cultivation, a reasonable amount of water and fertilizer is vital for the promotion of fruit growth and the production of stable and high yields. Current research on blueberries has focused mainly on agronomic traits [13], cultivation techniques [14], introduction and breeding [15], and processing [16]. Few studies have explored the impact of water and fertilizer coupling on blueberries [17]. Most earlier studies offered no technical guidance for blueberry production to Chinese fruit farmers and there was no attempt made to optimize fertilizer and water coupling for maximizing blueberry bush growth.

The present study aimed to determine the optimal water and fertilizer coupling regime for high fruit yield and quality, providing a theoretical basis for water and fertilizer management in blueberry production in the Guizhou region. The findings of this study will provide some practical technical guidance to fruit farmers to enhance blueberry cultivation, help conserve water and fertilizers, and sustainably increase fruit yield and quality.

## 2 Materials and methods

### 2.1 Test site

These trials were conducted in Xuanwei Town, Majiang County, Guizhou Province, China, located at 26°21′–26°31′ N and 107°33′–107°47′ E in a sub-tropical monsoon humid zone with a warm and humid climate year-round. The annual average temperature, rainfall, sunshine hours, and frost-free period were 15.70°C, 1266 mm, 1200 h, and 293 days, respectively. The base soil was an acid yellow type with pH 4.35–5.50, organic matter 23.90 g kg$^{-1}$, total nitrogen (N) 0.784 g kg$^{-1}$, total phosphorus (P) 0.19 g kg$^{-1}$, and total potassium (K) 2.8 g kg$^{-1}$.

### 2.2 Experimental design

Five-year-old rabbiteye blueberry (*Vaccinium virgatum).* 'Britewell' shrubs were used as test material. The two main variables were irrigation water level (W) and fertilizer application rate (F). In this study, twelve water-fertilizer coupling treatments were compared taking no application of water and fertilizer as control. Details of treatments used is given in Table 1. The fertilizers, made of chemically pure compounds, were ammonium sulfate (N content 21.2%), superphosphate (P$_2$O$_5$ content 60.6%), and potassium sulfate (K$_2$O content 63.2%) were used. Experiment was laid out following randomized complete block design (RCBD) with three replications. Each replication has three plants with plant-to-plant distance of 1.5 m. Fertilization and irrigation were conducted four times a year as follows: early March (before flowering), early May (before fruit production), late August to early September (after fruit production, flower bud differentiation period), and early December (reducing fertilizer). Ditches that were 50-cm long, 20-cm wide, and 20-cm deep were dug on the left and right sides of the outer periphery of the canopy projection. The fertilizer was dissolved in water and poured into the ditches, which were then covered with soil. In all other aspects, the blueberry plants were maintained according to standard cultivation practices. Preliminary trials indicated that the treatments used here would have significantly different effects on blueberry growth [18].

### 2.3 Determination of indices

**2.3.1 Photosynthesis index.** In mid- to late July, on sunny or cloudless days, representative leaf blades with consistent growth trends were measured using an LI-6400 Portable

**Table 1. Fertilizer–water coupling experimental design (single application).**

| Fertilization amount (g plant$^{-1}$) | Irrigation volume (L plant$^{-1}$) | | | |
|---|---|---|---|---|
| (NH$_4$)$_2$SO$_4$ + Ca(H$_2$PO$_4$)$_2$ + K$_2$SO$_4$ | 1.25 (W1) | 2.50 (W2) | 3.75 (W3) | 5 (W4) |
| 29 + 5 + 10 (F1) | F1W1 | F1W2 | F1W3 | F1W4 |
| 59 + 10 + 20 (F2) | F2W1 | F2W2 | F2W3 | F2W4 |
| 118 + 20 + 40 (F3) | F3W1 | F3W2 | F3W3 | F3W4 |
| 0 (CK) | 0 | | | |

Note: Single fertilization and irrigation application rates are shown. F1 is low fertilizer, F2 medium fertilizer, F3 high fertilizer, W1 is low water, W2 is medium water, W3 is medium-high water, and W4 is high water.

Photosynthetic System Analyzer (Beijing Ligaotai Technology Co. Ltd., Beijing, China). The control flow rate and the leaf surface temperature were 500 μmol s$^{-1}$ and 125°C, respectively. Before measurement, leaf photosynthesis was induced for 5 min using 02B-LED red and LI-6400XT blue light sources (Beijing Ligaotai Technology Co. Ltd., China). The photosynthetically active radiation (PAR) gradient was 2000, 1800, 1600, 1400, 1200, 1000, 800, 600, 400, 200, 100, 60, 30, 15, and 0 μmol·m$^{-2}$·s$^{-1}$, which was set using an automated measurement program. The maximum and minimum waiting times were 200 s and 150 s, respectively. After every three treatments, the net photosynthetic rate ($P_n$), stomatal conductance ($G_s$), intercellular carbon dioxide concentration ($C_i$), and transpiration rate ($T_r$) were measured. The leaf blade water use efficiency (WUE) was calculated as follows [19]:

$$\text{WUE} = P_n/T_r \tag{1}$$

**2.3.2 Leaf physiological activity index.** Mature blueberry leaves were collected for leaf physiological activity analysis. The veins were excised and the laminae were ground and mixed. Chlorophyll content was determined after extraction with 80% acetone solution, and the soluble protein content was determined using the Coomassie Brilliant Blue-G250 staining method. The anthrone method (colorimetry) was used to determine soluble sugars and the nitro blue tetrazolium photochemical reduction method was used to determine superoxide dismutase (SOD) activity. Malondialdehyde (MDA) activity was determined using the thiobarbituric acid method.

Aforementioned traits (chlorophyll contents, SOD activity, and MDA activity) were determined using previously reported protocols by Mahawar et al. [20], Dehghan et al. [21] and Zhu et al. [22].

**2.3.3 Tree index.** The new shoot growth index was determined in May. Annual shoots were randomly selected to observe the growth of all new shoots. Flower-bud differentiation was observed mainly in January by counting the number of all flower buds.

**2.3.4 Membership function.** In this study, 12 water-fertilizer coupling treatments were compared taking no application of water and fertilizer as control; thus making 13 treatments in total (Table 1). Each treatment had 3 replications (3 × 13 = 39) and comprehensive evaluation with membership function by transforming the data related to physiological and growth responses in following Eqs (2) and (3).

$$X = (x - x_{min})/(x_{max} - x_{min}), \tag{2}$$

$$X = 1 - (x - x_{min})/(x_{max} - x_{min}), \tag{3}$$

In the above equations, X and x represent the coded and the average calculated value of each treatment, respectively; $x_{min}$ and $x_{max}$ represent the minimum and maximum value, respectively, obtained from each parameter from different treatments. The membership function values and average value were accumulated and calculated. The larger average value represented the optimal treatment group, as recommended by previous studies [23].

**2.3.5 Cluster analysis.** Systematic clustering was used to combine the physiological leaf and bush growth indicators under various water and fertilizer combinations using SPSS v. 25 (IBM Corp., Armonk, NY, USA).

## 2.4 Data processing

Excel v. 2010 (Microsoft Corp., Redmond, WA, USA) and SPSS v. 25 (IBM Corp., USA) program was used to analyze the data using one-way analysis of variance (ANOVA). Duncan's new multiple range test was used to separate means where ANOVA indicated significant differences at p≤0.05.

# 3 Results

## 3.1 Effects of water and fertilizer coupling on blueberry photosynthetic characteristics

All photosynthetic parameters (Pn, Ci, Tr) were improved under water and fertilizer application as compared with control treatment. Data revealed that, $P_n$ was relatively higher for F3W4, F2W3, and F1W3 and comparatively lower for F2W1 and F1W4 compared to other treatments. Combining low or medium fertilizer with medium-high water, or high fertilizer with high water increased the net foliar photosynthetic rate. $G_s$ was highest under the F3W4 treatment and lowest under the F2W4 treatment. For all water–fertilizer combinations, $C_i$ was higher compared to that in CK, reaching the highest value under F2W1 and the lowest under F2W4. Thus, the coupling effect of water and fertilizer increased $C_i$, which in turn enhanced foliar $P_n$. $T_r$ was relatively higher for F3W3, F3W4, F2W1, and F1W1 and comparatively lower for F1W4 and F2W4 compared to other treatments. Foliar WUE significantly differed among the various water–fertilizer combinations. Leaf WUE was highest in F2W4, followed by that in F2W3, F3W2, F1W4, F1W3, and F2W2; the lowest values were obtained under F2W1 (Table 2).

## 3.2 Effects of water and fertilizer coupling on the physiological activity in blueberry leaves

Compared to control, overall, the water and fertilizer coupling treatment increased soluble protein, soluble sugar, SOD, chlorophyll; decreased MDA. Compared with the control, F3W3 promoted the accumulation of soluble protein, while F1W3 and F1W4 decreased the soluble protein content of leaves.

Thus, the treatments with low fertilizer coupled with medium to high water were not conducive to soluble protein accumulation in the blueberry leaves. The foliar soluble sugar levels

**Table 2. Effects of water and fertilizer coupling on blueberry photosynthetic characteristics.**

| Treatments | $P_n$ (μmol m$^{-2}$ s$^{-1}$) | $G_s$ (mol m$^{-2}$ s$^{-1}$) | $C_i$ (μmol mol$^{-1}$) | $T_r$ (mmol m$^{-2}$ s$^{-1}$) | WUE (μmol mmol$^{-1}$) |
|---|---|---|---|---|---|
| F1W1 | 6.12 ± 1.00ab | 0.05 ± 0.00e | 281.46 ± 20.51a-c | 2.85 ± 0.20c | 2.15 ± 0.24c-e |
| F1W2 | 5.76 ± 1.06ab | 0.09 ± 0.01c | 297.29 ± 18.13a-c | 2.09 ± 0.24de | 2.76 ± 0.42c-e |
| F1W3 | 6.68 ± 1.17ab | 0.11 ± 0.00b | 299.23 ± 15.24a-c | 1.85 ± 0.07ef | 3.61 ± 0.57cd |
| F1W4 | 4.32 ± 0.74ab | 0.07 ± 0.00d | 291.43 ± 15.47a-c | 1.04 ± 0.04gh | 4.15 ± 0.64c |
| F2W1 | 3.76 ± 0.68b | 0.12 ± 0.00b | 339.03 ± 9.59a | 3.39 ± 0.12b | 1.11 ± 0.18e |
| F2W2 | 5.25 ± 0.95ab | 0.07 ± 0.01d | 287.48 ± 15.96a-c | 1.46 ± 0.15fg | 3.60 ± 0.54cd |
| F2W3 | 6.69 ± 1.12ab | 0.04 ± 0.00f | 277.16 ± 23.95a-c | 1.56 ± 0.11f | 4.29 ± 0.61c |
| F2W4 | 6.57 ± 1.16ab | 0.02 ± 0.00g | 266.26 ± 29.77bc | 0.72 ± 0.08h | 8.52 ± 1.55a |
| F3W1 | 6.5 ± 1.14ab | 0.11 ± 0.01b | 298.34 ± 14.67a-c | 2.37 ± 0.16d | 2.74 ± 0.40c-e |
| F3W2 | 6.54 ± 1.17ab | 0.03 ± 0.00f | 299.38 ± 23.28a-c | 1.53 ± 0.16f | 4.27 ± 0.65c |
| F3W3 | 5.78 ± 1.05ab | 0.11 ± 0.01b | 312.39 ± 14.45ab | 3.6 ± 0.25b | 1.62 ± 0.29de |
| F3W4 | 7.54 ± 1.39a | 0.17 ± 0.00a | 316.68 ± 13.32ab | 4.29 ± 0.15a | 1.77 ± 0.31de |
| CK | 5.68 ± 0.90ab | 0.03 ± 0.00f | 248.69 ± 22.49c | 0.94 ± 0.15h | 6.04 ± 1.10b |

Different lowercase letters indicate significant differences between the treatments (P < 0.05).

F1 = Low fertilizer [29 + 5 +10 g plant$^{-1}$ (NH$_4$)$_2$SO$_4$ + Ca(H$_2$PO$_4$)$_2$ + K$_2$SO$_4$, respectively]; F2 = Medium fertilizer [59 + 10 +20 g plant$^{-1}$ (NH$_4$)$_2$SO$_4$ + Ca(H$_2$PO$_4$)$_2$ + K$_2$SO$_4$, respectively]; F3 = High fertilizer [118 + 20 +40 g plant$^{-1}$ (NH$_4$)$_2$SO$_4$ + Ca(H$_2$PO$_4$)$_2$ + K$_2$SO$_4$, respectively]; W1 = Low water (1.25 L plant$^{-1}$); W2 = Medium water (2.50 L plant$^{-1}$); W3 = Medium-high water (3.75 L plant$^{-1}$); W4 = High water (5.00 L plant$^{-1}$); WUE = Water use efficiency, net photosynthetic rate ($P_n$), stomatal conductance ($G_s$), intercellular carbon dioxide concentration ($C_i$), and transpiration rate ($T_r$).

**Table 3. Effects of different water and fertilizer treatments on the physiological activity of blueberry leaves.**

|  | Soluble protein (%) | Soluble sugar (%) | SOD ($\mu$g g$^{-1}$) | MDA (nmol g$^{-1}$) | Chlorophyll (mg g$^{-1}$) |
|---|---|---|---|---|---|
| F1W1 | 6.22 ± 0.08b | 2.85 ± 0.09de | 113.82 ± 5.23d-f | 44.73 ± 1.62de | 1.43 ± 0.04c-e |
| F1W2 | 4.79 ± 0.10e | 3.17 ± 0.08bc | 119.89 ± 3.15de | 55.01 ± 2.72b-d | 1.37 ± 0.02e |
| F1W3 | 4.53 ± 0.22e | 3.57 ± 0.10a | 173.52 ± 3.32ab | 53.33 ± 2.87b-e | 1.39 ± 0.02e |
| F1W4 | 4.47 ± 0.10e | 2.76 ± 0.03e | 99.94 ± 2.24f | 65.94 ± 1.95a | 1.40 ± 0.01c-e |
| F2W1 | 5.15 ± 0.05d | 3.13 ± 0.038bc | 128.22 ± 8.53cd | 54.32 ± 2.55b-d | 1.58 ± 0.05b |
| F2W2 | 6.13 ± 0.04b | 3.26 ± 0.03b | 173.29 ± 2.68ab | 50.24 ± 2.99c-e | 1.75 ± 0.04a |
| F2W3 | 6.34 ± 0.05b | 3.58 ± 0.04a | 178.17 ± 6.78a | 43.65 ± 3.54e | 1.49 ± 0.02b-e |
| F2W4 | 6.17 ± 0.02b | 3.07 ± 0.07bc | 141 ± 3.13c | 63.21 ± 4.20ab | 1.41 ± 0.04c-e |
| F3W1 | 5.65 ± 0.07c | 3.04 ± 0.06cd | 109.17 ± 4.26ef | 51.4 ± 3.23c-e | 1.54 ± 0.04bc |
| F3W2 | 6.33 ± 0.05b | 3.06 ± 0.03bc | 124.82 ± 4.08c-e | 52.6 ± 4.66c-e | 1.75 ± 0.05a |
| F3W3 | 6.73 ± 0.17a | 3.16 ± 0.04bc | 157.26 ± 6.11b | 57.72 ± 3.31a-c | 1.45 ± 0.05c-e |
| F3W4 | 5.47 ± 0.07c | 2.8 ± 0.08e | 138.21 ± 7.21c | 59.31 ± 3.47a-c | 1.53 ± 0.05b-d |
| CK | 4.65 ± 0.11e | 2.85 ± 0.09de | 110.38 ± 8.99ef | 67.41 ± 3.13a | 1.18 ± 0.06f |

F1 = Low fertilizer [29 + 5 +10 g plant$^{-1}$ (NH$_4$)$_2$SO$_4$ + Ca(H$_2$PO$_4$)$_2$ + K$_2$SO$_4$, respectively]; F2 = Medium fertilizer [59 + 10 +20 g plant$^{-1}$ (NH$_4$)$_2$SO$_4$ + Ca(H$_2$PO$_4$)$_2$ + K$_2$SO$_4$, respectively]; F3 = High fertilizer [118 + 20 +40 g plant$^{-1}$ (NH$_4$)$_2$SO$_4$ + Ca(H$_2$PO$_4$)$_2$ + K$_2$SO$_4$, respectively]; W1 = Low water (1.25 L plant$^{-1}$); W2 = Medium water (2.50 L plant$^{-1}$); W3 = Medium-high water (3.75 L plant$^{-1}$); W4 = High water (5.00 L plant$^{-1}$); SOD, superoxide dismutase; MDA, malondialdehyde.

were highest under F2W2, F2W3, and F1W3 but lowest under F3W4 and F1W4. Therefore, medium fertilizer–medium/medium-high irrigation treatments and low fertilizer–medium-high water treatment promoted soluble sugar accumulation in blueberry leaves. The leaf SOD levels were highest under F2W2, F2W3, and F1W3 and lowest under F3W1 and F1W4. Hence, the medium fertilizer–medium/medium-high irrigation treatments and low fertilizer–medium-high water treatment supported foliar SOD accumulation and increased the antioxidant capacity of leaves. Leaf MDA levels were increased in control treatment under missing fertilizer and water application while other treatment combinations have decreased leaf MDA level. The low/medium fertilizer–high water treatments were most conducive to leaf growth. The foliar chlorophyll levels were higher under F2W1, F2W2, and F3W2 and lowest under F1W2 and F1W3. Thus, an appropriate fertilization rate may increase the relative chlorophyll content and improve the photosynthetic capacity of leaves (Table 3).

## 3.3 Effects of water and fertilizer coupling on blueberry bush growth

Compared to CK, all water–fertilizer treatments significantly increased the number of flower buds and new shoots (P < 0.05). The number of flower buds was highest under F2W2 and F2W3 and lowest under F3W1 and F1W4. Therefore, the medium fertilizer–medium/medium-high water treatments increased the number of flower buds. The number of new shoots was highest under F2W3, F2W2, and F3W2 and lowest under F3W1 and F1W4, indicating that medium fertilizer coupled with medium or medium-high water increased the number of new shoots (Table 4).

## 3.4 Membership function analysis

The average membership function values were higher for all water–fertilizer treatments compared to that of CK (Table 5). Therefore, the comprehensive indices of blueberry leaf physiological activity and growth under each water–fertilizer treatment were superior to those for CK. The mean membership function values for F2W2 and F2W3 were significantly higher than those for all other treatments (P < 0.05), indicating that the F2W2 and F2W3

**Table 4. Effects of different water and fertilizer treatments on the growth of blueberry flower buds.**

|  | Number of flower buds | Number of new shoots |
|---|---|---|
| F1W1 | 821.33 ± 37.33ef | 44.67 ± 1.76d-f |
| F1W2 | 965.33 ± 54.34a-c | 58.33 ± 7.8bc |
| F1W3 | 872 ± 21.39cdef | 68 ± 1.53ab |
| F1W4 | 790.33 ± 23.38ef | 48.67 ± 1.76c-e |
| F2W1 | 990.67 ± 38.46ab | 56 ± 2.31c |
| F2W2 | 1,046.67 ± 35.88a | 74 ± 1.15a |
| F2W3 | 1,050.67 ± 48.64a | 75.67 ± 2.33a |
| F2W4 | 902.33 ± 21.99b-e | 55.33 ± 5.04cd |
| F3W1 | 780.33 ± 21.73f | 47.33 ± 2.40c-e |
| F3W2 | 994 ± 44.38ab | 51.33 ± 1.76c-e |
| F3W3 | 943 ± 33.56a-d | 69.33 ± 4.67a |
| F3W4 | 838.67 ± 43.96d-f | 34.67 ± 1.76f |
| CK | 630.33 ± 32.52g | 41.33 ± 3.18ef |

F1 = Low fertilizer [29 + 5 +10 g plant$^{-1}$ $(NH_4)_2SO_4$ + $Ca(H_2PO_4)_2$ + $K_2SO_4$, respectively]; F2 = Medium fertilizer [59 + 10 +20 g plant$^{-1}$ $(NH_4)_2SO_4$ + $Ca(H_2PO_4)_2$ + $K_2SO_4$, respectively]; F3 = High fertilizer [118 + 20 +40 g plant$^{-1}$ $(NH_4)_2SO_4$ + $Ca(H_2PO_4)_2$ + $K_2SO_4$, respectively]; W1 = Low water (1.25 L plant$^{-1}$); W2 = Medium water (2.50 L plant$^{-1}$); W3 = Medium-high water (3.75 L plant$^{-1}$); W4 = High water (5.00 L plant$^{-1}$).

combinations were the most suitable for rabbiteye blueberry bush growth, followed by F3W3. F3W1. CK had the lowest average membership function values. For this reason, neither F3W1 nor F1W4 was conducive to rabbiteye blueberry bush growth.

## 3.5 Cluster analysis

The cluster analysis resolved the 13 treatments into four categories in the range of 4–10. F2W2, F2W3, and F3W3 were combined into one category, while F1W4 and CK were in

**Table 5. Membership values and comprehensive rankings.**

|  | SP | SS | SOD | MDA | Chl | flower buds | new shoots | average value | total ranking |
|---|---|---|---|---|---|---|---|---|---|
| F1W1 | 0.75 | 0.17 | 0.18 | 0.75 | 0.47 | 0.42 | 0.28 | 0.43 | 10 |
| F1W2 | 0.23 | 0.47 | 0.25 | 0.51 | 0.38 | 0.68 | 0.57 | 0.44 | 9 |
| F1W3 | 0.14 | 0.83 | 0.82 | 0.46 | 0.41 | 0.51 | 0.78 | 0.56 | 6 |
| F1W4 | 0.12 | 0.09 | 0.03 | 0.83 | 0.43 | 0.36 | 0.06 | 0.27 | 12 |
| F2W1 | 0.36 | 0.82 | 0.34 | 0.49 | 0.67 | 0.72 | 0.52 | 0.56 | 5 |
| F2W2 | 0.7 | 0.52 | 0.78 | 0.36 | 0.88 | 0.82 | 0.9 | 0.71 | 2 |
| F2W3 | 0.79 | 0.85 | 0.89 | 0.19 | 0.54 | 0.83 | 0.92 | 0.72 | 1 |
| F2W4 | 0.73 | 0.43 | 0.47 | 0.21 | 0.46 | 0.56 | 0.51 | 0.48 | 7 |
| F3W1 | 0.55 | 0.36 | 0.13 | 0.41 | 0.61 | 0.34 | 0.33 | 0.39 | 11 |
| F3W2 | 0.79 | 0.42 | 0.3 | 0.44 | 0.88 | 0.73 | 0.42 | 0.57 | 4 |
| F3W3 | 0.94 | 0.46 | 0.65 | 0.59 | 0.49 | 0.64 | 0.81 | 0.65 | 3 |
| F3W4 | 0.48 | 0.13 | 0.44 | 0.64 | 0.59 | 0.45 | 0.36 | 0.44 | 8 |
| CK | 0.18 | 0.17 | 0.14 | 0.88 | 0.14 | 0.07 | 0.2 | 0.25 | 13 |

F1 = Low fertilizer [29 + 5 +10 g plant$^{-1}$ $(NH_4)_2SO_4$ + $Ca(H_2PO_4)_2$ + $K_2SO_4$, respectively]; F2 = Medium fertilizer [59 + 10 +20 g plant$^{-1}$ $(NH_4)_2SO_4$ + $Ca(H_2PO_4)_2$ + $K_2SO_4$, respectively]; F3 = High fertilizer [118 + 20 +40 g plant$^{-1}$ $(NH_4)_2SO_4$ + $Ca(H_2PO_4)_2$ + $K_2SO_4$, respectively]; W1 = Low water (1.25 L plant$^{-1}$); W2 = Medium water (2.50 L plant$^{-1}$); W3 = Medium-high water (3.75 L plant$^{-1}$); W4 = High water (5.00 L plant$^{-1}$). SP, soluble protein; SS, soluble sugar; Chl, chlorophyll; SOD, superoxide dismutase; MDA, malondialdehyde.

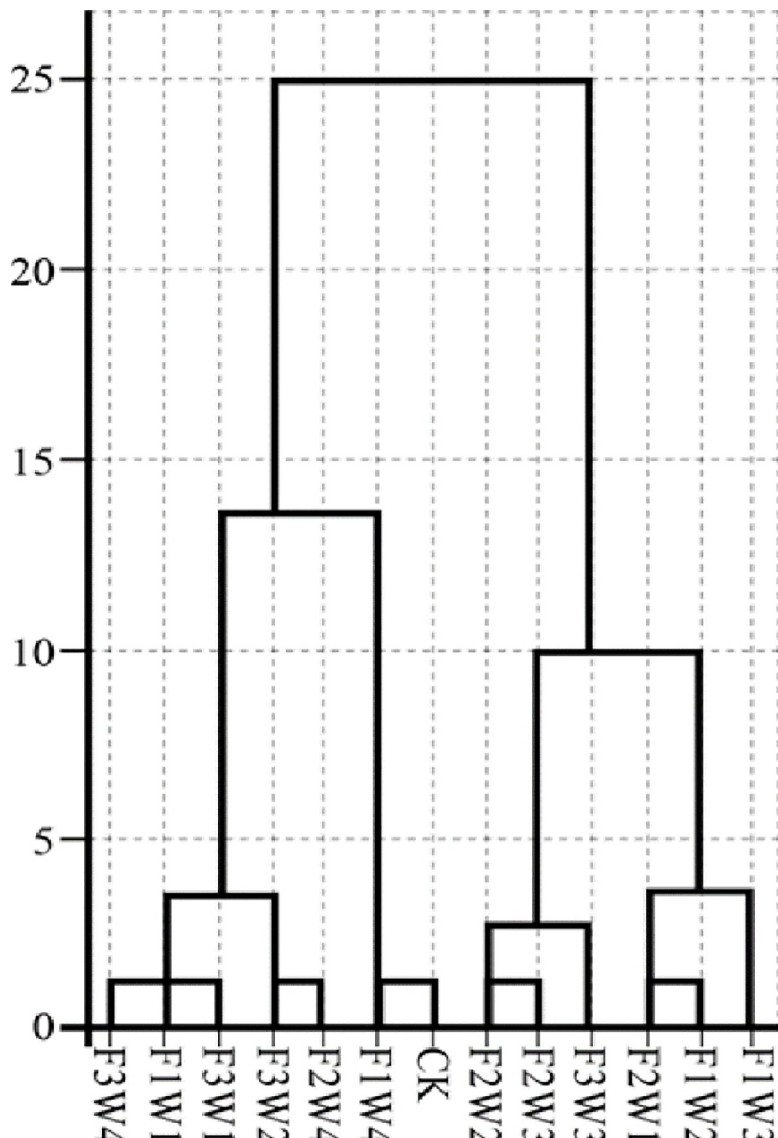

**Fig 1. Cluster analysis of various water and fertilizer combinations.** Treatment abbreviations are defined in Table 1.

another category. Thus, medium fertilizer with medium to medium-high water (F2W2 and F2W3) and high fertilizer and medium-high water (F3W3) provided the best water-fertilizer coupling, whereas low fertilizer combined with high water (F1W4) was the worst (Fig 1).

## 4 Discussion

### 4.1 Blueberry leaf physiology and bush growth characteristics under various water and fertilizer coupling regimes

Water and fertilizer dramatically influence plant growth, development, and the physiological characteristics of photosynthesis [24]. In response to various water and fertilizer coupling regimes, the $P_n$ for rabbiteye blueberry was in the range of 4.32–7.54 μmol m$^{-2}$ s$^{-1}$, which was higher than that of herbaceous strawberry (*Fragaria ananasa*) (5.27 μmol m$^{-2}$s$^{-1}$) [25], closing to raspberry shrubs (*Rubus microphyllus*) (8.8 μmol m$^{-2}$ s$^{-1}$) [26], while lower than that of

arbor walnut (*Jugla nsregia*)(20.31 μmol m$^{-2}$ s$^{-1}$) [27]. Here, we obtained similar trends in all treatments for $G_s$, $C_i$, and $T_r$. The photosynthetic capacities of various plant species differ in terms of their relative responses to water and fertilizer coupling. In general, the overall response of trees is stronger than those of shrubs and herbs, possibly owing to the greater photosynthetic capacity of trees and therefore higher net photosynthetic rate when compared to those of shrubs and herbs. In the present study, the highest $P_n$, $G_s$, and $T_r$ values were detected in blueberry leaves under the F3W4 treatment. Similar findings were reported for corn (*Zeay mays* L.) [28], strawberry (*Fragaria ananasa*) [25], spruce (*Picea glauca*) [29], and apple (*Malus domestica*) [30]. Probably because sufficient water and fertilizer are beneficial to the increase of leaf stomatal conductance, appropriate fertilizer application is beneficial to increase the chlorophyll content, thereby promoting the photosynthesis and transpiration of plants. This study found that there is a significant difference in WUE between different treatments, which might be due to the different effects of different water and fertilizer coupling modes on Pn, Tr and Gs (Table 2).

The effects of water and fertilizer on the soluble sugar, soluble protein, SOD, and chlorophyll content of blueberry leaves observed here were similar to those reported by Wassel et al. [31] for *Balady mandarin trees* (*Citrus reticulate*) and by El-Sayed et al. [32] for orange (*Citrus sinensis*). It shows that the interaction between water and fertilizer has the most significant effect on the physiological activities of plant leaves, followed by water. Here, the foliar MDA content was comparatively greater under high irrigation (W4). Aganchich et al. [4] reported similar findings for the olive (*Olea europaea*), and Mo et al. [33] made similar observations for watermelon (*Citrullus lanatus*). Overwatering may lower foliar enzyme activity by inhibiting branch growth and leaf expansion. Consequently, the leaf area and leaf number are reduced and the leaves abscise. The present study shows that the chlorophyll and soluble sugar levels in blueberry were at their maxima under W3 irrigation. The overall chlorophyll and soluble sugar levels under W1 were higher than those under W4, indicating that rabbiteye blueberry endures underwatering better than it does overwatering. Water deficit stimulates the production of osmotic substances in blueberries and enhances drought tolerance. In contrast, water excess may dilute the osmoprotectant soluble sugars; Paltineanu et al. [12] reported a similar conclusion. In another study, Gu et al. [34] found that moderate drought stimulated chlorophyll biosynthesis in the lowbush blueberry 'Chibowa' and the northern highbush blueberry 'Ruika'; however, its production decreased with increasing drought stress. The results of the present study are consistent with those findings. An increase in chlorophyll biosynthesis advances photosynthesis and photosynthate accumulation, promoting overall plant growth and improving plant stress resistance. For this reason, the blueberry chlorophyll and soluble sugar levels were higher under W1 than under W4 treatments. The results of this study also demonstrated that the medium and medium-high water treatments coupled with medium fertilizer promoted flower bud and new shoot formation, whereas neither dehydration nor overwatering were conducive to flower bud or new shoot growth. Similar discoveries were reported for rabbiteye and Gao Cong bilberry [35,36]. Comparable to the study on grapes by Shi et al. [37], the interaction between water and fertilizer significantly affected new shoot growth in blueberry plants. Fertilization is crucial for the maintenance of a nutrient balance in plant and soil; the promotion of plant growth and high-quality fruit products. Chen et al. [38] and Zhou et al. [39] found that medium fertilization improves the photosynthetic capacity and chlorophyll, soluble sugar, and soluble protein contents of blueberry leaves. The right combination of N, P, and K fertilizer improves fruit weight, and secures nutritional balance making the bush robust. Its reported that nitrogen (N) is the main constituent element of chlorophyll. When the crop is lack of N, the chlorophyll content in the body will be decreased, along with the weakness of photosynthesis intensity and the reduction of photosynthetic product, so the

crop growth will be slow [40]; and appropriate amount of N and P fertilizer supply is conducive to the development of crop vegetative organs (roots, stems, leaves, etc.) and the coordinated development of plant population indicators, which can increase the growth of plants and stems, and at the same time can improve the slow and slow extension of crop leaves under water stress conditions. Adverse effects such as reduced leaf area increase the accumulation of crop dry matter and reduce yield loss [41]. The present study showed that the maximum SOD content, flower bud and new shoot numbers were obtained with moderate fertilizer; as earlier were reported by Chen et al. [38] for southern highbush blueberry 'O'Neal'. Therefore, moderate rate of fertilization seemed ideal for rabbiteye blueberry plant growth; exceeding this optimum fertilization rate is resource wastage.

## 4.2 Impact of various water and fertilizer coupling regimes on rabbiteye blueberry

Water and fertilizer coupling considerably influenced rabbiteye blueberry physiology and growth. Especially this research showed that F2W2, F2W3, and F3W3 represented the best water and fertilizer coupling regimes, whereas F1W4 was the worst. It is probably that the absorption, transportation and utilization of nutrients by plants all depend on soil moisture, and the soil moisture status determines to a large extent the reasonable amount of fertilizer, and in the process of cooperating water and fertilizer application, it is necessary to properly play the high-efficiency role of water. Too high fertilizer concentration can cause damage to crops and even burn seedlings, as well as ecological problems such as soil salinization. The absorption, transportation and utilization of nutrients by plants depend on soil moisture. The soil moisture status determines the reasonable amount of fertilizer to a large extent, and more moisture can make nutrient elements migrate to the root surface and absorb quickly. Therefore, reasonable water and fertilizer can promote the growth and development of plants [42]. This shows that the ratio of water and fertilizer is of great significance to the precise and operable planting of rabbit-eye blueberries. In the blueberry production area of Guizhou, fruit growers should use the F2W2, F2W3, and F3W3 regimes to boost rabbiteye blueberry production. In contrast, F1W4 is not conducive to rabbiteye blueberry growth and should be avoided.

The present study showed that F2W2 and F2W3 had the highest membership function values for soluble sugar, SOD content and the number of flower buds and new shoots. Either too much or too little water impedes rabbiteye blueberry bush growth. Water deficits and excess fertilizer might induce a potentially lethal nutrition excess state, whereas over watering, fertilizer deficiency and poor soil permeability restricts root growth. Previous studies confirmed that high soil water content displaces the $CO_2$ in the soil pores and hinders its diffusion. This in turn leads to a decline in soil permeability, respiration and microbial and root activity, ultimately inhibiting plant growth [43]. Inadequate fertilization does not meet plant nutrient requirements for normal growth, whereas over fertilization may induce a potentially fatal nutrition excess state in plants. Therefore, carefully calculated and planned fertilization will help maximize crop productivity and quality, while mitigating soil and groundwater pollution.

This study focused only on rabbiteye blueberry cultivated in the Guizhou production area of China and established the ideal water and fertilizer coupling regime only for variety under study. However, other blueberry varieties, including 'Neil', 'Nagao O'Neill', 'Blue Rain', and 'Bei Gao Cong Jersey', are also cultivated in Guizhou. The effects of water–fertilizer coupling may vary among different blueberry varieties. Thus, field trials should be conducted in the Guizhou region to determine the optimal regime for each blueberry variety. This will be the next important research task of the team.

## 5 Conclusion

The interaction between water and fertilizer had the strongest effect on the physiological characteristics and growth of rabbiteye blueberry, followed by water and then fertilizer. Appropriate water and fertilizer coupling regimes may improve the photosynthetic efficiency of blueberry plants and promote growth. Comprehensive membership function and cluster analyses showed that medium fertilizer coupled with medium to medium-high irrigation or high fertilizer coupled with medium-high irrigation were the optimal regimes for rabbiteye blueberry cultivation in Guizhou.

## Supporting information

**S1 Data.**
(XLS)

## Author Contributions

**Data curation:** Xiaolan Guo, Delu Wang, Zongsheng Huang, Khuram Mubeen, Muhammad Shakeel.

**Investigation:** Xiaolan Guo.

**Methodology:** Xiaolan Guo, Shuangshuang Li.

**Project administration:** Delu Wang.

**Resources:** Delu Wang.

**Software:** Xiaolan Guo, Shuangshuang Li, Delu Wang, Zongsheng Huang.

**Supervision:** Delu Wang.

**Validation:** Xiaolan Guo, Shuangshuang Li, Muhammad Shakeel.

**Visualization:** Shuangshuang Li, Zongsheng Huang.

**Writing – original draft:** Xiaolan Guo.

**Writing – review & editing:** Naeem Sarwar, Khuram Mubeen, Mubshar Hussain.

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
