## [Decision Letter · Decision Letter 0]

4 Feb 2021

PONE-D-21-00083

Effects of Water and Fertilizer Coupling on the Physiological Characteristics and Growth of Rabbiteye Blueberry

PLOS ONE

Dear Dr. Wang,

Thank you for submitting your manuscript to PLOS ONE. After careful consideration, we feel that it has merit but does not fully meet PLOS ONE’s publication criteria as it currently stands. Therefore, we invite you to submit a revised version of the manuscript that addresses the points raised during the review process.

We look forward to receiving your revised manuscript.

Kind regards,

Mohammad Golam Mostofa, PhD

Academic Editor

PLOS ONE

Journal Requirements:

5. Please ensure that you refer to Figure 1 in your text as, if accepted, production will need this reference to link the reader to the figure.

6. Please upload a copy of Figure 4, to which you refer in your text. If the figure is no longer to be included as part of the submission please remove all reference to it within the text.

7. We note you have included a table to which you do not refer in the text of your manuscript. Please ensure that you refer to Table 7 in your text; if accepted, production will need this reference to link the reader to the Table.

Reviewers' comments:

Reviewer's Responses to Questions

**Comments to the Author**

1. Is the manuscript technically sound, and do the data support the conclusions?

Reviewer #1: Partly

Reviewer #2: Yes

2. Has the statistical analysis been performed appropriately and rigorously? 

Reviewer #1: Yes

Reviewer #2: Yes

3. Have the authors made all data underlying the findings in their manuscript fully available?

Reviewer #1: Yes

Reviewer #2: Yes

4. Is the manuscript presented in an intelligible fashion and written in standard English?

Reviewer #1: No

Reviewer #2: Yes

5. Review Comments to the Author

Reviewer #1: This study searches for suitable combinations of water and fertilizers to improve the morpho-physiological and biochemical adaptations of Rabbiteye Blueberry. Blueberry is an emerging cosmopolitan fruit tree in recent years, has been promoted on a trial basis in various provinces and cities of Southern China including in Guizhou Province. For the growth and high fruit production of Rabbiteye Blueberry, the optimal water and fertilizer is very essential and authors have found some optimal water and fertilizers rate to improve growth of Rabbiteye Blueberry. However, I think the manuscript can be made much stronger if some major revision work is been done before its acceptance for publication in PLOS ONE.

1. The manuscript should be thoroughly revised with native English speaker. In many cases sentences have no logical flow and therefore it is difficult to find author's message.

Abstract

2. Line 32-36: In abstract author mentioned maximum level of irrigation water and fertilizer application improved photosynthetic capacity in line 32-34. Whereas, in line 34-35 author stated high fertilizer–medium water treatment increased Chlorophyll contents. Again, in line 35-36 they mentioned medium fertilizer–medium water treatment increased Chlorophyll contents, which is very confusing and contradictory with one another. Please correct it.

3. Line 34-36: Please indicate clearly which treatment combination increased leaf-soluble protein.

4. Line 40-42: Author should be more careful during writing chemical formula. Please check the whole manuscript to fix similar kind of error.

Introduction

5. Line 60: Please replace hm-2 with international unit throughout the manuscript.

Materials and Methods

6. Please clearly indicate plant to plant and line to line distance in this section.

7. Line 129-136: Authors have mentioned that they used 65º c dry sample to measure Chlorophyll content, SOD activity and MDA content using protocol of Mahawar et al. 2018, 2019 and Khator et al. 2020. However, it was found that above mentioned authors used fresh samples to measure these variables, which is contradictory with the sample used by authors and their reference. Give procedure in details.

8. Line 142-150: It is not clear that which parameters authors included under membership value and which parameters included under anti-membership value. As author recommend optimal water and fertilizer rate with the help of membership value therefore it is very important to know which parameters belongs to membership value and which parameters included under anti-membership value. However, author did not mention it clearly in this section. Author can follow Roy et al. (2021) Agricultural Water Management 246, 106712 to make this portion clearer for reader.

Results

9. It is suggested to revise the results portion to make a clear comparison of different treatments with control treatment.

10. Line 160-161: Make a clear comparison either water and fertilizer caused increased or decreased growth indexes compare to control.

11. Line 179-180: Again, I suggest to make a clear comparison either water and fertilizer caused increased or decreased growth indexes compare to control.

12. Line 182-183: Same as above.

13. Line 187-189: Increase of MDA contents indicates plants membrane damage then how increase of MDA content under low/medium fertilizer-high water could be conductive to leaf growth?

14. Line 217: Author mentioned that F1W4 represents the lowest membership function value however table 8 represented that CK had lowest membership function value. Please correct it

15. Line 224-225: No appropriate in here, suggested to delete.

Discussion

16. Please, avoid re-mentioning results repeatedly in the discussion. This was already mentioned in the Results section. What happens with the nutrients and the lack of interaction between the doses of N and P? Why does that happen? Please explain.

17. Line 232-233: What is the actual relation of this sentence with the current study?

18. Line 235-237: What is the actual relation of this sentence with the current study?

19. Line 237-240: Discussion needs to be focused on mechanism rather simply description of verification of results. What kind of strategy is being employed by selected plants and give appropriate mechanism why this happened?

20. Line 240-243: This sentence is not clear, please rephrase.

21. Line 248-250: Author cited a huge number of Chinese Journal reference; it is suggested to reduce this number and add appropriate International Journal reference.

22. Line 290-295: This portion is the simple description of the cluster analysis results and quite similar with L 220-224. Revise this and focus on key points results and discuss with appropriate mechanism.

Table 8

23. Line 467: Why gas-exchange parameters such as Pn, Tr, Ci, Gs and WUE were not included in the Membership Function? If author include these parameters calculate accordingly maybe the average membership value and ranking will be changed.

Reviewer #2: Overall comments

The manuscript entitled “Effects of Water and Fertilizer Coupling on the Physiological Characteristics and Growth of Rabbiteye Blueberry” seems to be well designed. In my opinion, the MS should be accepted for publication after addressing few simple issues.

Specific comments

(a)Author’s measures two oxidative stress parameters (MDA and SOD activity) however authors didn’t give any explanation why these two parameters were considered for investigation and their relationship with fertilizer and irrigation doses either in introduction or discussion section.

(b) Authors used plenty of common names of different plant species in the discussion section. Authors should use scientific name along with their common names viz., Cucumber (Cucumis sativus).

(c)Plenty of type-setting errors were found throughout the MS. Authors should follow the journal guidelines in writing reference in the reference section.

6. PLOS authors have the option to publish the peer review history of their article (what does this mean?). If published, this will include your full peer review and any attached files.

Reviewer #1: No

Reviewer #2: No

---

## [Author Response · Author response to Decision Letter 0]

26 Mar 2021

March 1, 2021

Mohammad Golam Mostofa, PhD

Academic Editor 

PLOS ONE

Re: Effects of Water and Fertilizer Coupling on the Physiological Characteristics and Growth of Rabbiteye Blueberry (PONE-D-21-00083R1)

Dear Mohammad Golam Mostofa, PhD

 I, on behalf of all co-authors thank you and reviewers for the detailed review and providing us the opportunity to revise and improve the above-cited manuscript. We have carefully revised our manuscript according to the suggestions/comments made by the reviewer (see below point-to-point reply). 

We hope that the revised manuscript will be accepted for publication in ‘PLOS ONE’. 

We wish you to hear from you in due course

Dr. Delu Wang

Corresponding author

Reviewers 1

1. The manuscript should be thoroughly revised with native English speaker. In many cases sentences have no logical flow and therefore it is difficult to find author's message.

Response：We thoroughly copyedit our manuscript for language usage, spelling, and grammar, and we employed a professional scientific editing service

2. Line 32-36: In abstract author mentioned maximum level of irrigation water and fertilizer application improved photosynthetic capacity in line 32-34. Whereas, in line 34-35 author stated high fertilizer–medium water treatment increased Chlorophyll contents. Again, in line 35-36 they mentioned medium fertilizer–medium water treatment increased Chlorophyll contents, which is very confusing and contradictory with one another. Please correct it.

Response: Said lines are revised now for more clarity. Please see Lines 35-37 of the revision.

3. Line 34-36: Please indicate clearly which treatment combination increased leaf-soluble protein.

Response: Line 35. The high fertilizer–medium high water treatment increased leaf-soluble protein contents.

4. Line 40-42: Author should be more careful during writing chemical formula. Please check the whole manuscript to fix similar kind of error.

Response: All formulas are rechecked and revised to remove errors in full manuscript; see lines 41-44of the revision .

5. Line 60. Please replace hm-2 with international unit throughout the manuscript. 

Response: hm-2 changed to ha-1 in whole manuscript. 

6. Please clearly indicate plant to plant and line to line distance in this section.

Response: mentioned the distance of 1.5 meters within the plants. Please see Line 121 of the revision.

7. Line 129-136: Authors have mentioned that they used 65º c dry sample to measure Chlorophyll content, SOD activity and MDA content using protocol of Mahawar et al. 2018, 2019 and Khator et al. 2020. However, it was found that above mentioned authors used fresh samples to measure these variables, which is contradictory with the sample used by authors and their reference. Give procedure in details.

Response: Fristly, I'm very sorry, all the blueberry leaves were fresh. Corrections have been made in revised manuscript regarding chlorophyll contents, SOD and MDA. Please lines 148-156 of the revision.

8. Line 142-150: It is not clear that which parameters authors included under membership value and which parameters included under anti-membership value. As author recommend optimal water and fertilizer rate with the help of membership value therefore it is very important to know which parameters belongs to membership value and which parameters included under anti-membership value. However, author did not mention it clearly in this section. Author can follow Roy et al. (2021) Agricultural Water Management 246, 106712 to make this portion clearer for reader.

Response: Thank you reviewers for their comments. I have made changes here. The experiment was arranged using a varied irrigation and fertilizer under completely randomized. Within this design a total of four water doses were used: W1(1.25L, low), W2 (2.5 L, medium), W3 (3.75 L, medium high), W4 (5 L, highest). Furthermore, three N doses as:F1(F1=29g,low), F2(F2 =59g,medium), F3(F3= 118g,highest) were also supplied. With no fertilization and irrigation as the control, there are 13 treatments in total. Treatments using various regimes of W and F doses were selected for this study and presented in Table 1. Each treatment had 3 replications (3 × 13 = 63), and pots were randomly ordered inside the plastic shed. Interactive effects of W × F on integrated growth performance (IGP) of blueberry were obtained by transforming the data related to physiological and growth responses in following Eqs. (2) and (3), and the results are shown in Table 8.

Membership value = (X - Xmin)/(Xmax - Xmin), (2)

Anti-membership value = 1- (X - Xmin)/(Xmax - Xmin), (3)

In the above equation, X and x represent the coded and the average calculated value of each treatment, respectively; xmin and xmax represent the minimum and maximum value, respectively, obtained from each parameter from different treatments. To calculate IGP of blueberry, we used Eq ([Disp-formula pone.0254013.e002]) for those growth indices that were positively correlated with W, F application. However, Eq ([Disp-formula pone.0254013.e003]) was used for those growth indices that were negatively correlated with W,F application, as recommended by previous studies. [23].

Results

9. It is suggested to revise the results portion to make a clear comparison of different treatments with control treatment.

Response: Thank you for the reviewer’s suggestion. We have made appropriate revision.

10. Line 160-161: Make a clear comparison either water and fertilizer caused increased or decreased growth indexes compare to control.

Response: comparison of growth parameters with control treatment have been added in revised manuscript. overall, the water and fertilizer coupling treatment increased Pn, Ci, Tr; decreased WUE.” Please see Line 191-194 of the revision.

11. Line 179-180: Again, I suggest to make a clear comparison either water and fertilizer caused increased or decreased growth indexes compare to control.

Response: Revised as per suggestion. overall the water and fertilizer coupling treatment increased soluble protein, soluble sugar, SOD,chlorophyll; decreased MDA. Compared with the control, F3W3 promoted the accumulation of soluble protein, while F1W3 and F1W4 decreased the soluble protein content of leaves.” Please see lines 205-208 of the revision.

12. Line 182-183: Same as above.

Response: Thank you for the reviewer’s suggestion. “Thus, the treatments with low fertilizer coupled with medium to high water were not conducive to soluble protein accumulation in the blueberry leaves.” changed to “Thus, F1W3 and F1W4 were not conducive to soluble protein accumulation in the blueberry leaves.” Please see lines 208-209of the revision.

13. Line 187-189: Increase of MDA contents indicates plants membrane damage then how increase of MDA content under low/medium fertilizer-high water could be conductive to leaf growth?

Response: Thank you for the reviewer’s suggestion. Thank you for pointing this out. According with your advice, we corrected the relevant part in manuscript. Please see line 216-218 of the revision, described it as “Leaf MDA levels were increased in control treatment under missing fertilizer and water application while other treatment combinations have decreased leaf MDA level. Therefore, the medium fertilizer–medium water treatments were most conducive to leaf growth’’.

14. Line 217: Author mentioned that F1W4 represents the lowest membership function value however table 8 represented that CK had lowest membership function value. Please correct it.

Response: Thank you for your comments. We have revised it “CK had the lowest average membership function values .”

15. Line 224-225: No appropriate in here, suggested to delete.

Response: OK, we deleted it.

16. Please, avoid re-mentioning results repeatedly in the discussion. This was already mentioned in the Results section. What happens with the nutrients and the lack of interaction between the doses of N and P? Why does that happen? Please explain.

Response: Thank you for the reviewer’s suggestion, we have made changes in Line 300-309 of the revision. 

17 And 18. Line 232-233 and lines 235-237: What is the actual relation of this sentence with the current study?

Response: Strawberries are herbs, raspberries are shrubs, and walnuts are trees. It is concluded that blueberries and raspberries are photosynthetically close. It shows that they have similar photosynthetic capacity with strawberries, raspberries and walnuts. We just want to show that the photosynthetic capacity of shrubs lies between trees and herbs; see lines 255-257 of the revision. 

19. Line 237-240: Discussion needs to be focused on mechanism rather simply description of verification of results. What kind of strategy is being employed by selected plants and give appropriate mechanism why this happened?

Response: It may be due to the fact that under suitable water conditions is beneficial to increase the stomatal conductance, under suitable conditions, adding fertilizers is beneficial to increase the chlorophyll content, or high water and high fertilizer are beneficial to improve the diversity and stability of soil microbial communities, and are beneficial to the soil ecological environment. Or because it helps to reduce the content of MDA in the leaves, thereby promoting the photosynthesis and transpiration of crops. The influence mechanism of water and fertilizer treatment on photosynthesis is relatively complicated, and further research is needed in the future.

20. Line 240-243: This sentence is not clear, please rephrase.

Response: Sentence is rephrased for more clarity. see lines 264-269 of the revision.

21. Line 248-250: Author cited a huge number of Chinese Journal reference; it is suggested to reduce this number and add appropriate International Journal reference.

Response: Chinese journal references are reduced and replaced with other studies published in some other international journals. 

22. Line 290-295: This portion is the simple description of the cluster analysis results and quite similar with L 220-224. Revise this and focus on key points results and discuss with appropriate mechanism.

Response: Thank you reviewers for their comments. Respective description is now added in revised version for explanation; see lines 316-329.

23. Line 467: Why gas-exchange parameters such as Pn, Tr, Ci, Gs and WUE were not included in the Membership Function? If author include these parameters calculate accordingly maybe the average membership value and ranking will be changed.

Response: We are agreed with reviewer suggestion. This time, leaf enzyme activity and growth indicators were measured in spring, while photosynthetic characters were measured in summer. In Guizhou, China, the spring and summer seasons vary greatly. In order to reduce the seasonal errors, photosynthetic indicators were not taken together to make Membership function analysis. But even if we add photosynthetic indicators, our best combination is still the middle fertilizer treatment, but the ranking of other treatments did change.

Reviewers 2

(a) Author’s measures two oxidative stress parameters (MDA and SOD activity) however authors didn’t give any explanation why these two parameters were considered for investigation and their relationship with fertilizer and irrigation doses either in introduction or discussion section.

Response: Thank you for the reviewer’s suggestion. I have added relative explanation in introduction; see lines 55-64.

(b) Authors used plenty of common names of different plant species in the discussion section. Authors should use scientific name along with their common names viz., Cucumber (Cucumis sativus).

Response: Thank you for the reviewer’s suggestion. Scientific names of all plant species at their 1st occurrence in whole manuscript.

(c) Plenty of type-setting errors were found throughout the MS. Authors should follow the journal guidelines in writing reference in the reference section.

Response: References are now written according to journal format and all type-setting errors are removed.

---

## [Decision Letter · Decision Letter 1]

3 May 2021

PONE-D-21-00083R1

Effects of Water and Fertilizer Coupling on the Physiological Characteristics and Growth of Rabbiteye Blueberry

PLOS ONE

Dear Dr. Wang,

Thank you for submitting your manuscript to PLOS ONE. After careful consideration, we feel that it has merit but does not fully meet PLOS ONE’s publication criteria as it currently stands. Therefore, we invite you to submit a revised version of the manuscript that addresses the points raised during the review process.

We look forward to receiving your revised manuscript.

Kind regards,

Shafaqat Ali

Academic Editor

PLOS ONE

Journal Requirements:

Additional Editor Comments (if provided):

Reviewers have now commented on your paper. One reviewer recommend minor revision and the other one accept. If you are prepared to undertake the work required, I would be pleased to reconsider your paper for publication.

Reviewers' comments:

Reviewer's Responses to Questions

**Comments to the Author**

1. If the authors have adequately addressed your comments raised in a previous round of review and you feel that this manuscript is now acceptable for publication, you may indicate that here to bypass the “Comments to the Author” section, enter your conflict of interest statement in the “Confidential to Editor” section, and submit your "Accept" recommendation.

Reviewer #1: (No Response)

Reviewer #2: All comments have been addressed

2. Is the manuscript technically sound, and do the data support the conclusions?

Reviewer #1: Yes

Reviewer #2: Yes

3. Has the statistical analysis been performed appropriately and rigorously? 

Reviewer #1: Yes

Reviewer #2: Yes

4. Have the authors made all data underlying the findings in their manuscript fully available?

Reviewer #1: Yes

Reviewer #2: Yes

5. Is the manuscript presented in an intelligible fashion and written in standard English?

Reviewer #1: Yes

Reviewer #2: Yes

6. Review Comments to the Author

Reviewer #1: Comments to the Author

This is a much-improved version of a paper I have seen before and the authors have addressed all the issues raised. However, I still feel it needs some works before it is publishable.

L163-166: I think that this experiment was not conducted in a plastic shed and no pots also used in here. Moreover, author used membership function not integrated growth performance (IGP) in their analysis. Therefore, it is suggested to revise these sentences according to the experimental design.

L190-191: "and, by extension" -What is the actual meaning of this?

L256-257: Please mention the scientific name for strawberry and spruce also. Please the whole manuscript to fix such error.

L65: Please mention the scientific name of orange in here.

L293-294: Sentence is not clear please revise it.

L318: Sentence is not clear.

L417: Author should be more careful during writing the names of the authors name in the reference section. For example: in 23 number reference the author’s name will be Mostofa MG not Mostfa MG.

Reviewer #2: The authors revised the MS according to my previous comments. The authors also amend the corrections of other reviewers.

7. PLOS authors have the option to publish the peer review history of their article (what does this mean?). If published, this will include your full peer review and any attached files.

Reviewer #1: **Yes: **Rana Roy, PhD

Department of Agroforestry & Environmental Science

Sylhet Agricultural University

Sylhet 3100, Bangladesh.

Reviewer #2: **Yes: **Professor Dr Mohammad Anwar Hossain

---

## [Author Response · Author response to Decision Letter 1]

31 May 2021

We have consider all the suggestions and revised the manuscript as per reviewer comments. Response sheet is uploaded in separate file

---

## [Decision Letter · Decision Letter 2]

18 Jun 2021

Effects of Water and Fertilizer Coupling on the Physiological Characteristics and Growth of Rabbiteye Blueberry

PONE-D-21-00083R2

Dear Dr. Wang,

We’re pleased to inform you that your manuscript has been judged scientifically suitable for publication and will be formally accepted for publication once it meets all outstanding technical requirements.

Kind regards,

Shafaqat Ali

Academic Editor

PLOS ONE

Additional Editor Comments (optional):

As both reviewers recommend your MS for publication. I am delighted to accept your manuscript for publication

Reviewers' comments:

Reviewer's Responses to Questions

**Comments to the Author**

1. If the authors have adequately addressed your comments raised in a previous round of review and you feel that this manuscript is now acceptable for publication, you may indicate that here to bypass the “Comments to the Author” section, enter your conflict of interest statement in the “Confidential to Editor” section, and submit your "Accept" recommendation.

Reviewer #1: All comments have been addressed

Reviewer #2: All comments have been addressed

2. Is the manuscript technically sound, and do the data support the conclusions?

Reviewer #1: Yes

Reviewer #2: Yes

3. Has the statistical analysis been performed appropriately and rigorously? 

Reviewer #1: Yes

Reviewer #2: Yes

4. Have the authors made all data underlying the findings in their manuscript fully available?

Reviewer #1: Yes

Reviewer #2: Yes

5. Is the manuscript presented in an intelligible fashion and written in standard English?

Reviewer #1: Yes

Reviewer #2: Yes

6. Review Comments to the Author

Reviewer #1: The authors have addressed all the issues raised and now the manuscript can be accepted for publication.

Reviewer #2: The authors revised the MS based on reviewers comments. In my opinion the MS should be accepted for publication.

7. PLOS authors have the option to publish the peer review history of their article (what does this mean?). If published, this will include your full peer review and any attached files.

Reviewer #1: **Yes: **Dr. Rana Roy, Department of Agroforestry & Environmental Science, Sylhet Agricultural University, Sylhet 3100, Bangladesh.

Reviewer #2: **Yes: **Professor Mohammad Anwar Hossain

---

## [Editor Report · Acceptance letter]

25 Jun 2021

PONE-D-21-00083R2 

Effects of Water and Fertilizer Coupling on the Physiological Characteristics and Growth of Rabbiteye Blueberry 

Dear Dr. Wang:

I'm pleased to inform you that your manuscript has been deemed suitable for publication in PLOS ONE. Congratulations! Your manuscript is now with our production department. 

Kind regards, 

on behalf of

Dr. Shafaqat Ali 

Academic Editor

PLOS ONE